# SOMAscan Proteomics Identifies Novel Plasma Proteins in Amyotrophic Lateral Sclerosis Patients

**DOI:** 10.3390/ijms24031899

**Published:** 2023-01-18

**Authors:** Elena Berrone, Giovanna Chiorino, Francesca Guana, Valerio Benedetti, Claudia Palmitessa, Marina Gallo, Andrea Calvo, Federico Casale, Umberto Manera, Alessandra Favole, Paola Crociara, Camilla Testori, Valerio Carta, Carlotta Tessarolo, Antonio D’Angelo, Giovanni De Marco, Maria Caramelli, Adriano Chiò, Cristina Casalone, Cristiano Corona

**Affiliations:** 1S.C. Neuroscienze, Istituto Zooprofilattico Sperimentale del Piemonte, Liguria e Valle d’Aosta, 10154 Turin, Italy; 2Cancer Genomics Laboratory, Fondazione Edo ed Elvo Tempia, 13900 Biella, Italy; 3Rita Levi Montalcini Department of Neuroscience, University of Turin, 10126 Turin, Italy; 4Neurology, Hospital Department of Neuroscience and Mental Health, Città della Salute e della Scienza Hospital of Turin, 10126 Turin, Italy; 5ASL TO4, 10034 Chivasso, Italy; 6Department of Veterinary Science, University of Turin, 10095 Grugliasco, Italy

**Keywords:** ALS, SOMAscan, plasma proteins, TARC, TIMP-3, nidogen

## Abstract

Amyotrophic lateral sclerosis (ALS) is a complex disease characterized by the interplay of genetic and environmental factors for which, despite decades of intense research, diagnosis remains rather delayed, and most therapeutic options fail. Therefore, unravelling other potential pathogenetic mechanisms and searching for reliable markers are high priorities. In the present study, we employ the SOMAscan assay, an aptamer-based proteomic technology, to determine the circulating proteomic profile of ALS patients. The expression levels of ~1300 proteins were assessed in plasma, and 42 proteins with statistically significant differential expression between ALS patients and healthy controls were identified. Among these, four were upregulated proteins, Thymus- and activation-regulated chemokine, metalloproteinase inhibitor 3 and nidogen 1 and 2 were selected and validated by enzyme-linked immunosorbent assays in an overlapping cohort of patients. Following statistical analyses, different expression patterns of these proteins were observed in the familial and sporadic ALS patients. The proteins identified in this study might provide insight into ALS pathogenesis and represent potential candidates to develop novel targeted therapies.

## 1. Introduction

Amyotrophic lateral sclerosis (ALS) is a fatal progressive neurodegenerative disease (ND) caused by a loss of upper and lower motor neurons that leads to death within 2–5 years [1,2,3]. ALS generally starts between 50 and 70 years old, and it is one of the most common motor neuron diseases (MNDs) among adults. The majority of cases are sporadic ALS (S-ALS) with no apparent familial history of the disease, while approximately 10% of cases are familial ALS (F-ALS) [4]. Historically, these forms are clinically and pathologically indistinguishable from one another, suggesting similar disease mechanisms [5].

However, recent studies have described ALS as a highly heterogeneous disease of unknown etiology [6]. This heterogeneity represents a strong limit for the formulation of an early diagnosis and the discovery of effective therapeutic treatments. ALS diagnosis is currently based on clinical assessment and electrophysiological examination, and usually, there is a significant delay of approximately 12 months between the onset of symptoms and the final diagnosis. In addition, as F-ALS is due to the presence of pathogenic variants in certain genes (outlined below), and these variants are also present in many S-ALS cases, currently, genetic analysis assists in patient stratification and precision medicine. Regarding treatment, the drugs approved thus far are only able to slow down the disease’s progression.

Although genetic biomarker research constantly finds new genes that are risk factors associated with ALS, many reviews deal with protein-based biomarkers for ALS diagnosis. Proteins are attractive because they directly affect phenotypes and can provide direct information on some biological pathways. Electrophoresis, mass spectrometry, the enzyme-linked immunosorbent assay (ELISA) and immunobead-based multiplex assays are the most utilized techniques for detecting and quantifying proteins in several neurodegenerative diseases, including ALS [7,8,9,10,11]. However, large-scale studies using these methods are often not feasible due to the limited numbers of samples or analytes that can be simultaneously studied or because of technical limitations in the quantification of low-abundance proteins.

ALS disease has usually been studied in the context of spinal cord diseases with candidate-based analyses of cerebrospinal fluid (CSF), assessing cell populations such as lymphocytic infiltrates and secreted factors of the central nervous system (CNS) [12]. Although CSF represents a good source of biomarkers for neurodegenerative disorders, as it is in direct contact with the extracellular space, blood represents a more useful substrate that is easily obtainable with minimally invasive methods [13]. Thousands of plasma proteins still remain unexplored for their relation to complex diseases such as ALS. Therefore, a systematic exploration of plasma proteins holds great potential for displaying novel biomarkers for diagnostic, prognostic and monitoring purposes. A promising alternative to current methods may be the validated high affinity-based protein technology, called SOMAscan (Multiplexed Proteomic technology SOMA logic Inc., Boulder, CO, USA), which is able to quantify multiple proteins in human biological liquids, including plasma [14,15,16,17,18,19,20,21].

The exact molecular pathways causing motor neuron degeneration remains unknown, but as for other NDs, they are likely to result from a complex interplay between multiple pathogenic cellular mechanisms which may not be mutually exclusive. These include genetic factors, excitotoxicity, oxidative stress, mitochondrial dysfunction, impaired axonal transport, neurofilament aggregation, protein aggregation and inflammatory and signaling pathway dysfunctions [22]. Moreover, the extremely heterogeneous phenotypic expression of ALS leads to the question of whether it is a single disease or a symptom resulting from distinct biological mechanisms. To answer this question, it is essential to use an innovative and global approach that allows determining the potential involvement of multiple signal pathways which are unexplored to date. A more complete understanding of these mechanisms is also required to support the development of new diagnostic and therapeutic approaches, as well as to identify an appropriate time window for therapeutic intervention.

The primary aim of this study is to identify proteins differentially expressed in ALS patients compared with controls via the SOMAscan assay. After bioinformatic analyses, four selected proteins are then independently validated using ELISA methods. The secondary aim is to analyze the different expression patterns of the selected proteins in F-ALS and S-ALS patients. 

## 2. Results

### 2.1. Patients Cohort Features

The demographic, genetic and clinical features of 47 patients [23] enrolled in the study are reported in Table 1.

Plasma samples from 16 of these patients (Cohort 1), including 8 F-ALS patients and 8 S-ALS patients, were enrolled in the exploratory proteomic analysis as analytical samples (female/male ratio: 8/8; mean age at disease onset and at sample collection: 62 and 64, respectively). Eight healthy subjects were selected as controls (female/male ratio: 3/5; average age at sample collection: 61).

The second cohort of 47 ALS patients and 32 age- and gender-matched controls were used to test plasma expression of the selected protein candidates using ELISA kits. The demographic characteristics among the ALS groups were comparable to those of the sampling in the control groups. In Cohort 2, the controls were divided into 19 neurologically healthy controls (H-CTR) and 13 controls with neurological pathologies other than MNDs (NoH-CTR).

Three ALS causative genes linked to over 50% of the F-ALS patients [24] were included in the validation cohort, namely 13% with the most common gene variant chromosome 9 open reading frame 72 (C9orf72), 9% superoxide dismutase 1 (SOD1) and 4% TAR DNA binding protein (TARDBP), while the remaining 35 patients were classified as S-ALS.

The mean progression rate (PR) or ALS functional rating scale revised (ALSFRS-R) slope (48-ALSFRS-R score at time of blood sampling/disease duration from disease onset to blood sampling in months) was 0.68 (SD 0.53) points lost per month. The ALS patients were stratified according to PR in slow progressors (PR ≤ 0.70 points lost per month) and fast progressors (PR > 0.70 points lost per month).

### 2.2. SOMAscan Proteomics Results

SOMAscan protein expression was evaluated in the plasma of Cohort 1’s patients and H-CTR, and an expression matrix with 24 columns (samples) and 1305 rows (proteins) was obtained. Unsupervised hierarchical clustering (HC) and PCA analysis were initially performed, but neither of them were able to separate H-CTR from ALS patients. Two-sample class comparison was then carried out either by significance analysis of microarrays (SAM) or by linear models for microarray analysis (limma). HC of the 24 samples using the deregulated proteins by both methods revealed some discrimination between ALS patients and H-CTR samples, as shown in Figure 1.

The class comparison results are also reported in Figure 2 as a volcano plot, with significantly upregulated and downregulated proteins by limma or SAM delimitated by the dashed lines and the orange dots, respectively.

Under SAM analysis, 42 proteins were identified as statistically significantly differently expressed between the patients and H-CTR. Among these, 8 proteins (TIMP-3, PDGF-BB, nidogen, NID 2, SH21A, TARC, Gro-b/g and PKC-D) were upregulated, while 34 (VCAM-1, ISLR2, NCAM-L1, Ephrin-A3, growth hormone receptor, ENPP7, TLR4:MD-2 complex, TF, BMPR1A, TLR4, HAI-1, sTie-1, MASP3, IL-13 Ra1, RET, RGMB, carbonic anhydrase I, IL-18 BPa, RGM-C, BCAM, kallikrein 7, ephrin-A2, IL-5 Ra, BST1, kallikrein 8, LYPD3, IR, nectin-like protein 2, IGF-I sR, CD23, sE-selectin, CYTT and TECK) were downregulated (Appendix A). Analysis with limma revealed eight proteins that were more expressed and nine that were less expressed in ALS patients than in H-CTR (Appendix A). A fractional cut-off for the fold change of the circulating protein content between cases and controls may be biologically meaningful, since we were dealing with proteins that are secreted or released within extracellular vesicles or can even result from apoptotic bodies. When evaluating the two typologies of cases in more detail, S-ALS or F-ALS versus H-CTR analyses revealed that 34 and 8 proteins were significantly modified, respectively (Appendix A) by the SAM method. When applying limma analysis, 21 proteins were differentially expressed in S-ALS versus H-CTR and 29 in F-ALS versus H-CTR (Appendix A).

Finally, the results of each group of analysis were compared using Venn diagrams (Figure 3a,b). The up- and downregulated proteins for both methods are visualized in bold. Two significantly upregulated proteins, thymus- and activation-regulated chemokine (TARC) and metalloproteinase inhibitor 3 (TIMP-3), were identified when comparing all ALS patients or only S-ALS versus H-CTR. Furthermore, SAM analysis revealed the upregulation of two isoforms of nidogen 1 and 2 (NID1 and NID2) in ALS and S-ALS patients versus H-CTR, respectively.

### 2.3. System Biology Analysis of Pathway Dysregulated in ALS Patients

Based on the upregulated and downregulated proteins in ALS patients versus H-CTR by either the SAM or limma methods, functional enrichment analysis was carried out using all the proteins assayed by Somascan or only the subset of secreted proteins as a background. Gene ontology (GO) lists were obtained, associating each protein with biological processes (BP), cellular components (CC), molecular functions (MF), Biocarta or Kegg pathway. Figure 4a,b details the overrepresented biological processes within the up- or downregulated proteins, respectively, using all the proteins assayed by Somascan as a background. The only significant cellular component overrepresented within the upregulated proteins was the basement membrane (BM) (GO:0005604).

Close examination of the GO annotation of the four upregulated proteins (TARC, TIMP 3, NID1 and NID2) identified by SOMAscan revealed their direct or indirect physiopathology functions on the BM (Appendix A).

When the subset of secreted protein was used as the background, we found only one statistically significant (*p*-value < 0.05) BP overrepresented within the downregulated proteins: GO:0050767~regulation of neurogenesis.

Finally, in order to further explore the probable role of the four differentially expressed proteins, their gene names were uploaded to Metacore^TM^ software and processed by the shortest-path algorithm. Through this analysis, it was possible to observe either direct interactions within protein pairs or links with one intermediary protein in the form of a directed network between these elements, viewed as nodes (Figure 5). A protein network involving TARC (also named C-C motif chemokine ligand 17 (CCL17)), TIMP-3, NID1 and NID2 as “central hubs” (red circle) was also developed. Matrix metalloproteinases (MMPs), particularly MMP-9 and MMP-13, were involved in the network through potential interactions with three of the uploaded proteins (CCL17, TIMP-3 and NID1). Moreover, the shortest-path algorithm highlighted the involvement of other intermediary molecules, such as stromelysin-1, CCL5, p53 and amyloid beta. In Table 2, the types of interaction and related PubMed references are described.

### 2.4. Validation of TARC, TIMP 3, NID1 and NID2 Using ELISA

TARC, TIMP 3, NID1 and NID2 were selected as putative biomarker candidates due to their statistical significance in the exploratory proteomics study and their biological relevance.

To confirm the preliminary SOMAscan findings, a validation cohort (Cohort 2) was analyzed by commercial ELISA kits. First, the ALS patients were compared with H-CTR and NoH-CTR by a Wilkoxon rank sum test, showing that TARC and TIMP3 were significantly higher in the patients compared with the neurologically healthy controls (*p* < 0.05) (Figure 6a,c), while no significant differences were present for NID1 and NID2 (Figure 6e,g). Next, the ALS patients were divided, based on clinical category, into F-ALS and S-ALS, and the two groups were compared to the controls (H-CTR and NoH-CTR). TARC was significantly higher at baseline in the S-ALS and in F-ALS groups (Figure 6b). Similarly, TIMP3 was higher in the S-ALS group compared with the neurologically healthy controls (*p* < 0.05) but not in the F-ALS group (Figure 6d). By comparing only F-ALS with all control groups, NID2 was significantly increased in the familial patients (*p* < 0.002) (Figure 6h). No differences were seen for the NID1 plasma levels in either category (Figure 6f).

Subsequently, receiver operating characteristic (ROC) curves were generated for each protein, and the respective area under the ROC curve (AUC) values were calculated. As presented in Figure 7a and Appendix A, the NID2 ROC curve showed the most relevant AUC value (0.815, *p* < 0.0001) in discriminating F-ALS patients versus CTR. Both the TARC and TIMP-3 ROC curves in S-ALS patients versus CTR had AUC values of 0.67 (*p* < 0.010) (Figure 7b and Appendix A). Finally, only the TARC protein showed a significant AUC value (0.68, *p* < 0.005) (Figure 7c and Appendix A) for ALS patients. In conclusion, according to Swets classification [47], only NID2 quantification appeared to be moderately accurate in discriminating F-ALS versus the controls.

### 2.5. Different Expression Patterns of Plasma Proteins in F-ALS and S-ALS Patients

Stepwise multivariable logistic regression analysis was then used to identify the protein combinations associated with the presence of ALS disease. The combination of TARC with NID1 resulted in moderate accuracy in discriminating ALS and S-ALS patients from the controls, showing AUCs of 0.75 and 0.74, respectively (Figure 8a,b). The association of NID2 with TARC and TIMP3 yielded an AUC value of 0.94 (Figure 8c) in predicting F-ALS, resulting in a very high clinical impact. Moreover, as presented in Appendix A, the moderately differentiated ALS group had significantly increased TARC and NID1 expression compared with the controls (*p* < 0.05). Similarly, TARC expression was increased in the moderately differentiated S-ALS group (*p* < 0.0024). The combination of NID2 and TIMP3 was significantly higher in the well-differentiated F-ALS group compared with the controls (*p* < 0.05).

## 3. Discussion

Under SOMAscan proteomic analysis of the plasma samples from 16 ALS patients and 8 healthy controls, 42 circulating proteins with altered expression in ALS cases were identified. Furthermore, the expression of plasma proteins in a heterogeneous ALS patient group were compared, including subjects with three ALS causative mutations linked to over 50% of the F-ALS group.

Blood plasma was chosen since it is considered a highly accessible source and an attractive body fluid for biomarker development. It is increasingly accepted that blood, as a connective tissue, potentially contains evidence of all processes occurring within the organism, at least in trace amounts [48]. Although ALS is a disease of the CNS, protein could move from damaged cells into the interstitial fluid and then to the blood via the brain’s lymphatic system [49,50], and thus several studies have investigated ALS-related protein changes in blood [51,52,53,54,55,56,57]. However, the wide dynamic range of plasma proteins and the presence of high-abundance proteins such as albumin could be key barriers for performing discovery proteomics [58]. To overcome these problems, there have been advances in technologies that allow depletion of high-abundance proteins. However, the degree to which relevant low-abundance proteins are lost during processing to remove high-abundance proteins is unclear and could be highly variable [59]. As demonstrated in several studies, a SOMAscan assay was able to analyze and quantify all expected 1305 proteins in the original plasma samples without pretreatment.

Based on their statistical and biological significance, four proteins were selected and validated in the plasma of a partially independent cohort of 47 ALS patients and 32 controls. The TARC, TIMP-3, NID1 and NID2 protein levels were significantly increased in the plasma of ALS patients. The ROC curves showed TARC as the most interesting biomarker candidate, together with NID1 or TIMP3 and NID2. Our results demonstrated that these proteins have a significant association with ALS disease and a different expression pattern in F- and S-ALS patients.

TARC, also named CCL17, is a powerful chemokine produced in the thymus and by antigen-presenting cells such as dendritic cells, macrophages and monocytes, which are associated with macrophage and microglial polarization. Chemokines are involved in neuroinflammation, and their role in several inflammatory diseases of the CNS has been investigated [60,61,62,63,64]. Regarding ALS, several members of the chemokine family were found to be altered in the CSF and blood of patients compared with the controls [65,66,67,68]. However, similar to other chemokines and cytokines, the ability of TARC to induce inflammation in various NDs is known [38,39,40,41], while little information is available on its role in ALS. In one study, a significant increase in TARC was observed in the sera of ALS patients [69], and recently, in a clinical trial, TARC expression was monitored for its possible role in controlling cytopathic microglial activation during ALS progression [70]. Therefore, the upregulation of plasma TARC observed in the present study could suggest an important and thus far unrecognized role as an inflammatory biomarker of ALS disease. Furthermore, the different association of TARC with NID1 or NID2, depending on the ALS group, could provide a better characterization of the disease subtypes.

With regard to TIMP-3, this protein belongs to the tissue inhibitor of the MMP family, a group of peptidases involved in degradation of the extracellular matrix (ECM). Expression of this protein is induced in response to mitogenic stimulation. TIMP3 is unique among the four TIMPs due to its ECM-binding property as well as its broadest range of substrates, including all MMPs, several disintegrin and metalloproteinases (ADAMs) and ADAMs with thrombospondin motifs (ADAMTSs) [71]. In addition to its metalloproteinase-inhibitory function, TIMP3 can interact with proteins in the extracellular space, resulting in its multivarious functions [72]. TIMP3 inhibits several membrane-bound molecules with sheddase functions, such as MMP14, MMP3 and tumor necrosis factor (TNF)-alpha converting enzyme, indicating that it plays a central part or role in several important reactions, including cellular growth, cellular death and tissue repair [73]. The mechanisms of action by which TIMP3 contributes to the NDs are slowly starting to be elucidated. Recently, an increase in TIMP3 was reported in Alzheimer’s disease (AD) brains [74], while in the SOD1G93A mouse model of ALS, high levels of TIMP-3 were observed only in the early phase of the disease, reducing at later stages proportionally with the number of live motor neurons in the spinal cord [75]. This different expression during ALS progression was also confirmed in S-ALS patients, where significantly decreased TIMP-3 protein expression was detected in postmortem spinal cord tissues [76]. These observations support the hypothesis that TIMP3 is a neuronal apoptotic protein whose early activation could directly contribute to motor neuron death. Therefore, the upregulation of TIMP3 observed in the present study might be associated with an initial phase of the disease and could represent a possible early indicator of ALS.

Finally, NID-1 and NID-2 are two isoforms of nidogens, a family of secreted glycoproteins also known as entactins. They are essential components to the BM alongside other components such as type IV collagen, proteoglycans (heparan sulfate and glycosaminoglycans), laminin and fibronectin [77]. While NID1 was found to be ubiquitously expressed throughout the BM’s surrounding blood vessels, nerves, cardiocytes and myotubes, NID2 appeared to be mainly restricted to the nerves and vasculature [78,79]. Despite the knockout mouse models of NID1 or NID2 suggesting a compensatory mechanism of these two isoforms [80,81], specific functions of these isoforms in different tissues, such as the neuro-muscular system, have been described [82,83,84]. To date, how nidogens change in NDs, including ALS, is mostly unknown. In fact, only one study on AD describes a decrease in nidogen expression in cerebral amyloid angiopathy [85]. The functional significance of nidogens in ALS is probably unknown due to the mutual compensation of NID1 and NIDI2 in embryonic lethality of the double knockout mice [78]. The distinct protein expression of NID1 and NID2 in S-ALS and F-ALS, respectively, observed in our analysis could be, therefore, an interesting new focus for future research. Familial and sporadic ALS are clinically indistinguishable, and it has long been speculated that they may share elements of the same pathogenic pathway. To date, the most commonly used models of ALS are transgenic mice overexpressing mutant SOD1, especially SOD1-G93A [86]. A large number of potentially therapeutic agents have been screened using these mutant SOD1 transgenic mice, based at least in part on the assumption that the disease mechanisms in S-ALS are similar to those in F-ALS. However, this assumption may not be appropriate. Although motor neuron degeneration is a shared downstream event in all types of ALS, the upstream pathways are likely to be different between F-ALS and S-ALS. In fact, our results showed that S-ALS and F-ALS are characterized by slightly different protein expressions. More studies on these differences could be useful to better distinguish both forms of the disease and discover more effective therapeutic treatments by using models such as the new hSOD1G93A swine model, characterized by strong muscular involvement [87].

Our study has several notable strengths. The use of SOMAscan proteomics and in-depth bioinformatics analyses enabled the measurement of over 1300 proteins and the identification of circulating plasma proteins with possible correlations in the pathophysiology of ALS. Functional enrichment analysis highlighted key novel biological functions that appear to be associated with ALS and correlated with BM organization. As described above, TIMP-3, NID1 and NID2 are in fact directly located in the ECM with different roles [88,89]. TARC is a chemokine active in the extracellular space which plays a key role in thymocyte migration. Although the existence of an ECM–chemokine interplay has not yet been established, recent data propose that ECM and chemokines might act in combination to drive immune cell migration [90,91,92] during inflammation. Therefore, validation of these proteins’ interactions by other methods and determination of the effects on their signaling will shed new light on the role of BM in ALS.

However, our study also has some limitations. First, the ALS discovery cohort was relatively small. Second, the SOMAscan technology applied to a complex matrix such as plasma could lead to unspecific protein detection, since a single aptamer may have more than one target. In addition, protein structure alterations due to different reasons, such as genetic polymorphisms, posttranslational modifications, oligomerization or degradation, may unpredictably alter the binding affinity and quantification [93]. Despite these limitations, the four proteins identified were validated by immunoassays, an independent technique, in a much larger—even if not entirely independent—cohort. Third, the sample size of F-ALS patients was smaller than that for the S-ALS patients due to the rarity of the familial form of this disease. However, the F-ALS mutations included in this study are the ones most represented in the literature and appear sufficient to detect differences between S-ALS and F-ALS patients. To increase the power of this comparison, it might be useful to study these proteins in transgenic animal models of F-ALS [87,94].

## 4. Materials and Methods

### 4.1. Patient Cohorts

Forty-seven ALS patients, diagnosed according to the El Escorial Revisited criteria [23], who were confluent at ALS Turin Center (CRESLA) and included in the Piemonte and Valle d’Aosta epidemiological register for ALS (PARALS), were recruited in this study. Twelve patients had a family history of F-ALS with known disease-causing mutations: 4 SOD1, 2 TARDBP and 6 C9orf72. The remaining 35 were classified as S-ALS.

At first, we selected a group (Cohort 1) of 16 ALS patients (8 F-ALS and 8 S-ALS) and 8 healthy controls for SOMAscan analysis (Table 1). A second group (Cohort 2, including the cases in Cohort 1) of 47 ALS patients and 32 controls was analyzed for final validation. In Cohort 2, the controls were divided into 19 for H-CTR and 13 for NoH-CTR (Table 2).

The patients underwent a clinical, neurophysiological (electromyography and motor evoked potentials), neuroradiological and genetic assessment. Periodic clinical follow-ups were performed every 3 months. The patients were classified as follows based on the onset site: classic, upper motor neuron predominant, pseudopolyneuritic and bulbar [95]. In addition, healthy control subjects were recruited. The patients and controls were properly informed of all aspects of the study and had to provide signed consent. The protocols and procedures were approved by the relevant local ethical committee (“Comitato Etico Interaziendale AOU Città della Salute e della Scienza di Torino—AO Ordine Mauriziano—ASL Città di Torino”). The clinical characteristics are summarized in Table 1 and Table 2.

### 4.2. Sample Collection and Plasma Extraction

All subjects were required to fast for at least 2 h before collection. Blood samples were drawn by venipuncture into EDTA tubes and centrifuged at 2000× *g* for 10 min at 4 °C within approximately 2 h of collection. Plasma supernatant was collected, divided into aliquots and frozen at −80 °C until analysis.

### 4.3. SOMAscan Assay

Proteomic analysis of the plasma samples (200 uL) was performed by SomaLogic, Inc. (Boulder, CO, USA) using a SOMAscan platform that quantifies the presence of 1305 target human proteins (secreted proteins, extracellular domains and intracellular proteins) that belong to broad biological groups, including receptors, kinases, cytokines, proteases, growth factors, protease inhibitors, hormones and structural proteins [96,97]. Most of these proteins are involved in signal transduction pathways, stress response, immune processes, phosphorylation, proteolysis, cell adhesion, cell differentiation and intracellular transport.

Proteins were measured using a Slow Off-Rate Modified Aptamer (SOMAmer)-based capture array that uses chemically modified single-stranded DNA sequences capable of uniquely recognizing the individual proteins, transforming a protein signal to a nucleotide signal that can be quantified using relative florescence on microarrays. Each SOMAmer array was validated for its specificity, upper and lower limits of detection and intra- and interassay variability. Plasma dilutions (0.005%, 1% and 40%) were applied to capture low-, medium- and high-abundance proteins. Positive and negative controls were also positioned on the array to understand if the experimental procedure was performed correctly.

### 4.4. SOMAscan Data Analysis

After the hybridization step, the microarrays were washed and scanned using a specific scanner containing a laser that excites the fluorescence of the fluorochrome used in the labeling step. The amount of the signal emitted is directly proportional to the amount of the dye on the microarray spot. The scanner can measure this quantity and process a digital image that reconstructs the position of each signal on the microarray based on the spot of its origin. The images were then analyzed by image analysis software that generates a text file containing a summary of the information on the intensity of the pixels contained in the spot and in the background. This information was processed, and a final value was generated that summarized the expression level of each detectable protein on the microarray.

Quality control was performed at the sample and SOMAmer level and involved the use of control SOMAmer arrays on the microarray and calibration samples. Hybridization controls measured the sample-by-sample variation in hybridization, while the median signal over all SOMAmer arrays measured the technical variability. The scale factors of these two metrics were used for normalization across all samples with acceptance criteria of 0.4–2.5 based on historic trends. SOMAmer-by-SOMAmer calibration occurred through repeated measurement of the calibration samples. Historic values were used to generate a calibration scale factor the acceptance criterion, which was that 95% of SOMAmer arrays had to have a calibration scale factor within 0.4 of the median.

The SOMAscan measures were finally reported as relative fluorescence units (RFU) in a summary ADAT file.

To find proteins significantly associated with the disease compared with the controls, two complementary methods were applied. First, a parametric analysis method (limma) was used which made assumptions on the distribution of expression signals, starting from the ADAT files (readat R package). This method is generally used for gene expression analysis when thousands of transcripts are measured. A linear model is applied which combines t statistics with a Bayesian approach. In this analysis, proteins with an absolute logFoldChange higher than 0.58 (which corresponds to positive or negative fold changes of 1.5) and with a raw *p*-value less than 0.05 were selected.

The second methodology (SAM) is instead nonparametric, and it was first proposed by Tusher and Tibshirani [98] for the analysis of gene expression microarrays, but it can also be applied to protein expression data, as specified on the official website (https://tibshirani.su.domains/SAM, accessed on 4 January 2023). SAM assigns a score to each gene or protein on the basis of the change in expression between two classes relative to the standard deviation of repeated measurements. By adding a small, strictly positive constant to the denominator of the usual t statistic, SAM solves the problem related to genes or proteins with low expression levels that have very small variance and may therefore have very large and unstable signal-to-noise ratios. For genes or proteins with scores greater than an adjustable threshold, SAM uses permutations of the repeated measurements to estimate the false discovery rate (fdr). In our analysis, the threshold was set at 0.7, and the selected proteins had fdrs not exceeding 0.15. Analyses were carried out using the siggens R package.

### 4.5. System Biology Analysis

To acquire new insights into potential pathophysiological pathways and biological functions underlying the ALS-related plasma protein signature, and to more precisely understand the complex interactions between the differentially expressed proteins and candidate upstream regulators, we performed functional enrichment analysis on the dysregulated proteins using the Database for Annotation, Visualization and Integrated Discovery (DAVID), which provides a comprehensive set of functional annotation tools for investigators to understand the biological meaning behind large lists of genes or proteins (https://david.ncifcrf.gov/, accessed on 15 December 2022). Two different backgrounds were used: all the proteins or only the secreted proteins, which were assayed on the Somascan platform. To retrieve the list of secreted proteins, the open-access SPRomeDB database (www.unimd.org/SPRomeDB, accessed on 15 December 2022), created by Chen G et al. [99], was used. The four validated ALS-associated proteins were included in further network analysis using the commercial knowledge Metacore^TM^ (GeneGo) database for protein–protein functional and physical interactions, the results of which were displayed as a functional network. Proteins without associations with any protein in the network were removed.

### 4.6. Enzyme-Linked Immunosorbent Assay

The selected candidate proteins (TARC, TIMP3, NID1 and NID2) were validated using sandwich enzyme immunoassays. The plasma protein levels were measured using commercial kits (human TARC ELISA Kit and human TIMP3 ELISA Kit—Abcam Inc., Toronto, ON, Canada, C; Human Nidogen-1 ELISA kit and Human Nidogen-2 ELISA kit—Thermo Fisher Scientific, Waltham, MA, USA) while following the manufacturer’s recommendations. The plasma was diluted to fall within the linear range of each respective assay.

### 4.7. Statistical Analysis

The SOMAscan proteomic data were reported in RFU. Quantile normalization and log transformation were performed for all RFU-reported data. Principal component analysis was performed to assess the presence of plate-specific effects. The ELISA results were analyzed using a Wilcoxon rank sum test, where *p* < 0.05 was considered to indicate a statistically significant difference. The diagnostic value of individual or combinations of plasma proteins was assessed by univariable logistic regression or multivariable stepwise logistic regression, respectively. The diagnostic accuracy was assessed by the AUC.

## 5. Conclusions

In conclusion, the identification of TARC, TIMP-3, NID1 and NID2 using SOMAscan combined with immunoassay validation placed these four proteins as possible biomarker candidates of ALS. Although clinical relevance of their upregulation in ALS requires functional validation and further investigation, the data of this study include a comprehensive analysis of the proteome, provide new insights into the biological pathways involved in ALS pathogenesis and could lead to the discovery of novel therapeutic targets in ALS.

## Figures and Tables

**Figure 1 ijms-24-01899-f001:**
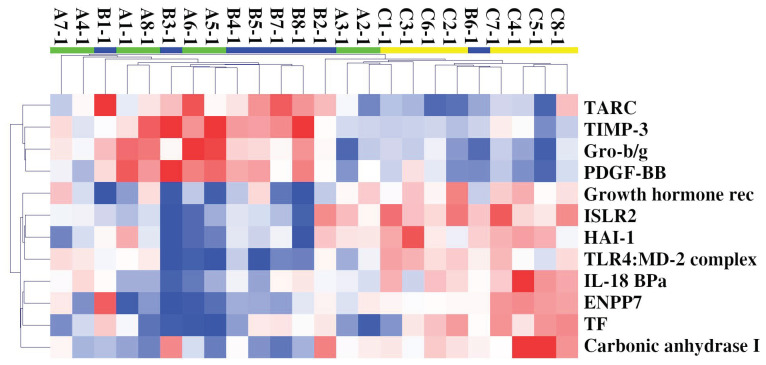
HC of 24 samples and 12 circulating deregulated proteins in ALS (F-ALS samples in green and S-ALS ones in blue) patients compared with H-CTR (in yellow). Pearson correlation was used as similarity metrics and the average as a linkage method. In the HC colormap, red denotes upregulation, and blue denotes downregulation. Each rectangle refers to the standardized expression across all samples, which is the expression level minus the row median divided by the row’s standard deviation. The relative color scheme uses the minimum (−2.0) and maximum (2.0) values in each row to convert values to colors. Abbreviations: Growth hormone rec = Growth hormone receptor.

**Figure 2 ijms-24-01899-f002:**
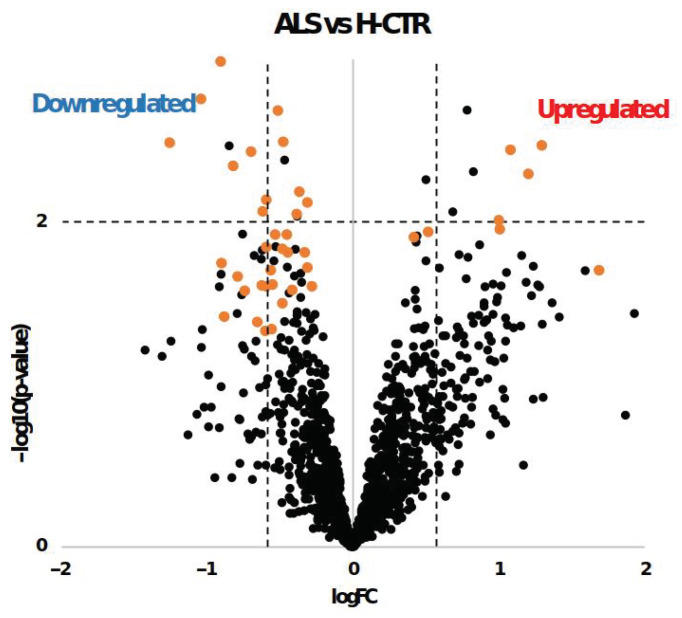
Volcano plot of log2 fold change (logFC) vs. log10 transformed *p*-values (−log10 (*p*-value)) for each protein in ALS patients compared with H-CTR. The x axis represents the logarithm (base 2) of the fold change between the two conditions. The fold change is the ratio between sample means of the two groups. Proteins with *p*-values less than 0.01 appear at the top of the plot above the horizontal dashed line. The vertical dashed lines indicate the thresholds of ±1.5-fold change. Significantly differentially expressed proteins obtained by SAM analysis are labeled with orange dots.

**Figure 3 ijms-24-01899-f003:**
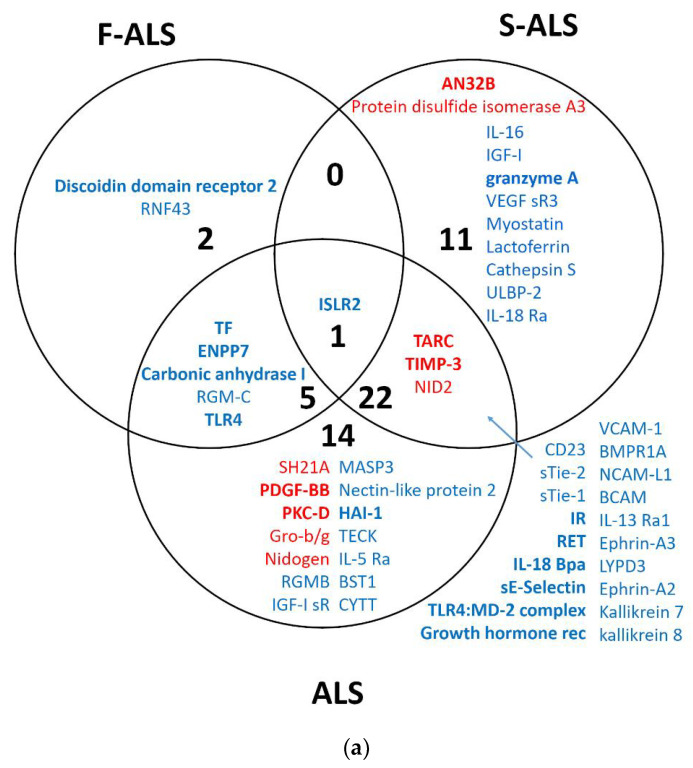
Venn diagram of overlapping proteins from a comparison of the proteomic results obtained for SAM (**a**) and limma (**b**) analysis. In bold are upregulated (red) or downregulated (blue) proteins identified by two statistical analysis methods.

**Figure 4 ijms-24-01899-f004:**
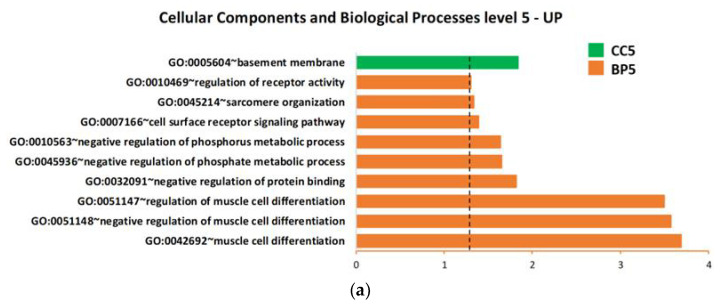
Biological process, cellular component and molecular function enrichment analysis of the lists of deregulated proteins in plasma of ALS patients compared with neurologically healthy controls, carried out using DAVID online tools on (**a**) upregulated proteins and (**b**) downregulated proteins. Enriched terms are plotted in ascending order of statistical significance within every functional category (*p*-value < 0.05). Values on the horizontal axis are −log10 (*p*-values). Abbreviations: BP = biological processes; CC = cellular components; MF = molecular functions; GO:0007157 heterophilic cell-cell adhesion via plasma membrane cell… = heterophilic cell-cell adhesion via plasma membrane cell adhesion molecules.

**Figure 5 ijms-24-01899-f005:**
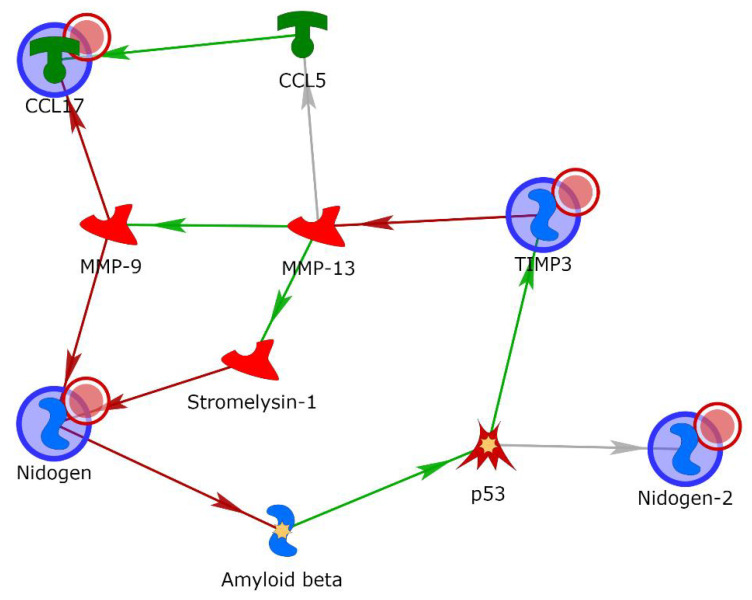
Gene network obtained with the shortest-path tool of Metacore^TM^ for the proteins TARC, TIMP-3, NID1 and NID2, allowing for two intermediary molecules. Green arrows indicate activation, while red indicates inhibition. Gray arrows mean that the type of interaction is not yet clear.

**Figure 6 ijms-24-01899-f006:**
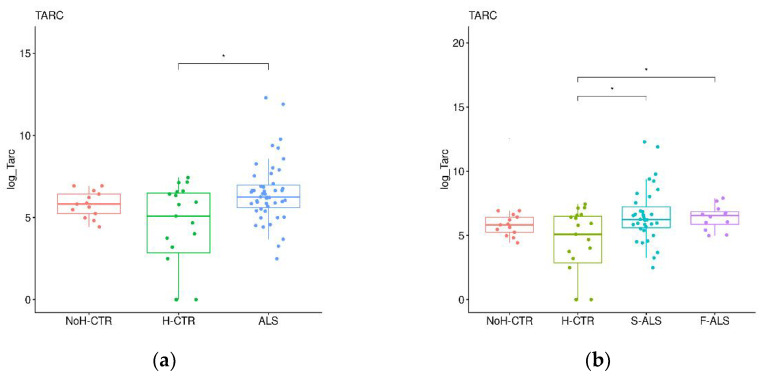
Box plots of individual plasma proteins levels in different diagnostic groups. The plasma TARC (**a**,**b**), TIMP-3 (**c**,**d**), NID1 (**e**,**f**) and NID2 (**g**,**h**) levels were measured in ALS patients (together on the left panels and divided into F-ALS and S-ALS in the right panels) and controls (H-CTR and NoH-CTR). * Post hoc *p*-value < 0.05.

**Figure 7 ijms-24-01899-f007:**
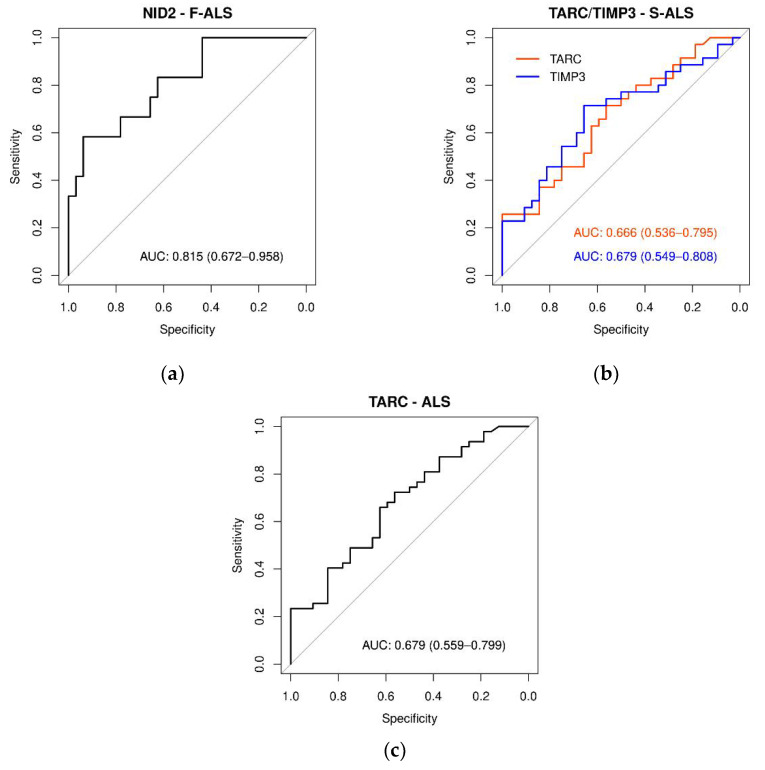
ROC curves of plasma protein levels in ALS, F-ALS and S-ALS referring to TARC, TIMP-3, NID1 and NID2 expression levels. The AUC indicates the diagnostic power: (**a**) NID2 in F-ALS patients (0.815), (**b**) TARC and TIMP3 in S-ALS patients (0.67) and (**c**) TARC in-ALS patients (0.68).

**Figure 8 ijms-24-01899-f008:**
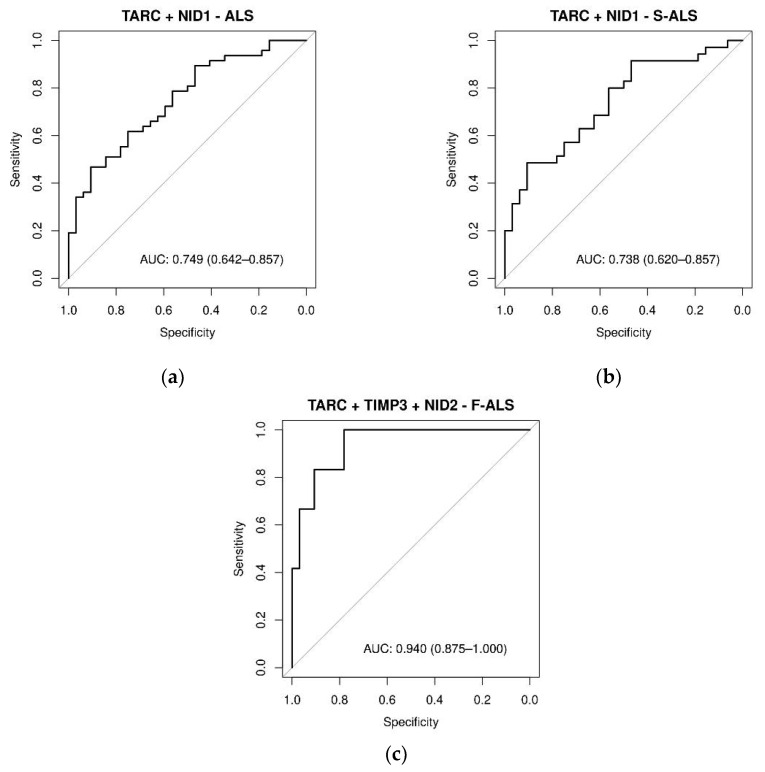
ROC curves of plasma proteins levels in ALS, F-ALS and S-ALS, referring to some significant linear combinations of TARC, TIMP-3, NID1 and NID2. The AUC indicate the diagnostic power: (**a**) TARC + NID1 in ALS patients (0.75), (**b**) TARC + NID1 in S-ALS patients (0.74) and (**c**) TARC + TIMP-3 + NID2 in F-ALS patients (0.94). The diagnostic model was constructed by using stepwise logistic regression analysis.

**Table 1 ijms-24-01899-t001:** Characteristics of control subjects and ALS patients in Cohort 1 for SOMAscan assay and Cohort 2 for protein validation by ELISA. Abbreviations: CTR = control group; H-CTR = healthy control; NoH-CTR = no healthy control; SC = sample collection; y = years; SD = standard deviation; F-ALS = familial ALS patients; C9orf72 = chromosome 9 open reading frame 72; SOD1 = superoxide dismutase 1; TARDBP = TAR DNA binding protein; S-ALS = sporadic ALS patients; FTD-ALS = frontotemporal dementia; ALSci = ALS with cognitive impairment; ALSbi = ALS with behavioral impairment; DNT = data not tested; DNP = data not provided; COPD = chronic obstructive pulmonary disease.

Cohort 1	
**CTR Group**	**8**
*Demographic Data*	
Female/male	3/5
Age at SC (y), mean (SD)	61 (12)
*Clinical Data*	
H-CTR (%)	8 (100)
NoH-CTR (%)	0 (0)
**ALS Group**	**16**
*Demographic Data*	
Female/male	8/8
Age at SC (y), mean (SD)	64 (11)
Age at onset (y), mean (SD)	62 (13)
*Genetic Data*	
F-ALS (%)	8 (50)
SOD1 (%)	3 (19)
C9orf72 (%)	3 (19)
TARDBP (%)	2 (12)
S-ALS (%)	8 (50)
*Clinical Data*	
ΔALSFRS (point decline per month), mean (SD)	0.68 (0.53)
Onset site:	
1. Classic (%)	5 (31)
2. Upper motor neuron predominant (%)	0 (0)
3. Pseudopolyneuritic (%)	5 (31)
4. Bulbar (%)	6 (38)
Cognitive status at diagnosis:	
Normal (%)	8 (50)
FTD (%)	1 (6)
ALSci (%)	1 (6)
ALSbi (%)	3 (19)
DNT (%)	3 (19)
Disease progression:	
Slow (%)	7 (44)
Fast (%)	9 (56)
Comorbidities:	
Hyperthension (%)	5 (31)
Diabetes (%)	1 (6)
Hypercolesteremia (%)	5 (31)
Hypothyroidism (%)	0 (0)
COPD (%)	2 (13)
Other comorbidity (%)	8 (50)
**Cohort 2**	
**CTR Group**	**32**
*Demographic Data*	
Female/male	12/20
Age at SC (y), mean (SD)	64 (11)
*Clinical Data*	
H-CTR (%)	19 (59)
NoH-CTR (%)	13 (41)
NoH-CTR pathology:	
Normal-pressure hydrocephalus (%)	8 (62)
Post-poliomyelitis syndrome (%)	2 (15)
Myasthenia gravis (%)	2 (15)
Parkinson’s disease (%)	1 (8)
**ALS Group**	**47**
*Demographic Data*	
Female/male	18/29
Age at SC (y), mean (SD)	65 (10)
Age at onset (y), mean (SD)	63 (11)
*Genetic Data*	
F-ALS (%):	12 (26)
SOD1 (%)	4 (9)
C9orf72 (%)	6 (13)
TARDBP (%)	2 (4)
S-ALS (%)	35 (74)
*Clinical Data*	
ΔALSFRS (points decline per month), mean (SD)	0.74 (0.60)
Onset site:	
1. Classic (%)	18 (38)
2. Upper motor neuron predominant (%)	3 (6)
3. Pseudopolyneuritic (%)	10 (11)
4. Bulbar (%)	14 (30)
DNP (%)	2 (4)
Cognitive status at diagnosis:	
Normal (%)	24 (51)
FTD (%)	4 (9)
ALSci (%)	9 (19)
ALSbi (%)	3 (6)
DNT (%)	7 (15)
Disease progression:	
Slow (%)	35 (75)
Fast (%)	10 (21)
DNP (%)	2 (4)
Comorbidities:	
Hyperthension (%)	20 (43)
Diabetes (%)	5 (11)
Hypercolesteremia (%)	5 (11)
Hypothyroidism (%)	3 (6)
COPD (%)	3 (6)
Other comorbidity (%)	11 (23)

**Table 2 ijms-24-01899-t002:** List of protein interactions and related PubMed references obtained by the database Metacore^TM^. Abbreviations: CCL5 = chemokine (C-C motif) ligand 5; CCL17 = C-C motif chemokine ligand 17 (TARC); MMP-13 = collagenase 3; MMP-9 = matrix metallopeptidase 9.

Network Object “FROM”	Object Type	Network Object “TO”	Object Type	Effect	Mechanism	Link Info	Ref.
Nidogen	Generic binding protein	Amyloid beta	Generic binding protein	Inhibition	Binding	Nidogen binds to and inhibits amyloid beta.	[25,26]
CCL5	Receptor ligand	CCL17	Receptor ligand	Activation	Binding	CCL5 binds to and activates CCL17.	[27]
Stromelysin-1	Metalloprotease	Nidogen	Generic binding protein	Inhibition	Cleavage	Stromelysin-1 cleaves and inhibits nidogen.	[28,29,30]
MMP-13	Metalloprotease	CCL5	Receptor ligand	Unspecified	Cleavage	MMP-13 cleaves CCL5.	[9]
MMP-13	Metalloprotease	Stromelysin-1	Metalloprotease	Activation	Binding	MMP-13 binds to and activates stromelysin-1.	[31]
p53	Transcription factor	TIMP3	Generic binding protein	Activation	Transcription regulation	p53 binds to gene TIMP3 promoter and increases its activity.	[32,33,34]
MMP-9	Metalloprotease	Nidogen	Generic binding protein	Inhibition	Cleavage	MMP-9 cleaves on Nidogen and inhibits its activity.	[28,30]
MMP-9	Metalloprotease	CCL17	Receptor ligand	Inhibition	Cleavage	MMP-9 cleaves on CCL17 and inhibits its activity.	[35]
p53	Transcription factor	Nidogen-2	Generic binding protein	Unspecified	Transcription regulation	By using chromatin immunoprecipitation with paired-end ditag sequencing analysis it was shown that nidogen 2 has binding sites for p53.	[36]
Amyloid beta 42	Generic binding protein	p53	Transcription factor	Activation	Transcription regulation	Amyloid beta 42 co-activates transcription of p53.	[37]
TIMP3	Generic binding protein	MMP-13	Metalloprotease	Inhibition	Binding	TIMP3 physically interacts with MMP-13 and decreases its activity.	[38,39,40]
MMP-13	Metalloprotease	MMP-9	Metalloprotease	Activation	Cleavage		[28,41,42,43,44,45,46]

## Data Availability

The data presented in this study are available within the article.

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
