# Peer review of "SOMAscan Proteomics Identifies Novel Plasma Proteins in Amyotrophic Lateral Sclerosis Patients"

_ijms, 2023, doi:10.3390/ijms24031899_

Round 1
Reviewer 1 Report
The study uses the aptamer-based SOMAscan platform to assess protein levels in the plasma of ALS patients compared to controls, and also between sporadic and familial subgroups.
I have major concerns relating to the statistical approach, explained below. Unfortunately the rest of the paper, including the choice of proteins for systems/pathway analysis and then for ELISA validation, is dependent on this initial statistical approach, which was flawed.
Considering that, for SOMAscan, there are likely no clear guidelines or substantial literature to draw upon in regards to which statistical test is the most appropriate (unlike in gene expression analysis where there are many comparative papers (e.g. https://journals.plos.org/plosone/article?id=10.1371/journal.pone.0012336, https://pubmed.ncbi.nlm.nih.gov/26057385/), it is understandable why the authors might wish to use two different algorithms (SAM and Limma) to identify which proteins are differentially expressed between groups. However, it is incorrect to then apply the criterion (as done in fig 1, and apparently fig 4, and elsewhere) that a protein should be considered a successful candidate if it is signficiant by either SAM or Limma - this is actually giving the weakest of both worlds: the approach is essentially giving each protein two opportunities to be significant (once with SAM and once with Limma), which invalidates the statistical assumptions of either approach.
If the authors wish to strengthen their conclusions by using these two separate algorithms then the criterion should be AND not OR. Proteins should be considered as suitable candidates if they are significant with both SAM and Limma. It might be excusable to maintain strict cut-offs on one of the two algorithms but allow weaker thresholds on the other (perhaps focusing on Limma as SAM has problems with low sample numbers, although I take the point that SAM is non-parametric).
In general, a lot more detail could be given on the statistical tests - e.g. version information for SAM, model used for Limma, clear descriptions of all thresholds/cut-offs applied for p-values and fold-changes.
Relating to this, why was a 1.5-fold change cut-off apparently applied for Limma but not for SAM? And is it biologically meaningfully to follow-up with pathway analysis on any protein whose level changes only fractionally? (even if, in this experiment, the levels happened to be very consistent)
More detail should be given of the enrichment testing in DAVID. What is the background universe of proteins? Is it all human proteins? Or only those of the SOMAscan platform? Or is it all those of the SOMAscan platform that are typically present in plasma? This choice will enormously affect the results.
It might be useful to provide rationale defending why it is even useful to conduct a systems biology analysis on a plasma sample. WHat are the limitations of such an analysis? What questions are truly being asked of the analysis, since systems ontologies are generally structured around cell physiology rather than secretory composition? A clear rationale around this should lead back to inform the decision of which background to include in enrichment testing.
The comparison of familial and sporadic seems of limited use - at least, more narrative could be included to better explain why the authors included this analysis, what they hoped to achieve from it, and what are the limitations of it (for example, to what extent is the F-ALS group dominated by one or other of the known causally mutated genes?).
As to figure 1, of course you will see discrimination on the heatmap when the heatmap includes only those proteins that are already identified for their ability to discriminate (i.e. are differntially expressed). It might be more informative to have clustering or PCA results *without* preselection of proteins - i.e. including all proteins assayed.
Figure 1 has no colour key for sample labelling or explanation of the A, B, C sample numbers
What is the up- or down- regulation value for a given sample and protein expressed relative to? Is it relative to the row average?
More commentary should be made on the potential limitations of SOMAscan (e.g. https://www.ncbi.nlm.nih.gov/pmc/articles/PMC6277005/)
Minor typos/suggestions (not exhaustive):
L28: decade > decades
L30: 'In this contest' > 'In pursuit of this'?
L31-32: the phrase 'proteomic background of ALS patients' lacks context - the meaning is not clear, and should be constrained to plasms: e.g. 'proteomic profile of ALS patient blood plasma'.
L47-48: "Most ALS cases (90%) are sporadic (S-ALS) with unknown etiology while approximately 10% of patients have a familial form of the disease (F-ALS)". I think the wording here is a little inaccurate. As the authors know, causal mutations are often identified in S-ALS patients, so etiology can sometimes be known. Conversely, not all cases of F-ALS need have a known mutation. The distinction between S-ALS and F-ALS is whether there are other cases in the immediate family.
L68: Removev'by'
L82: as well as to identify *an* appropriate..
L90: "an accurate bioinformatic": without further details, the word 'accurate' has no useful meaning here and should be removed
L163: "Both up- and down-regulated proteins were visualized in bold". I assume this is meant to say that up- and down-regulated proteins shared by SAM and Limma are indicated in bold.
Author Response
Please see the attachment and revisions in the manuscript.

Reviewer 2 Report
Study review: SOMAscan Proteomics Identifies Plasma Biomarkers with dif- 2 ferent expression patterns in familial and sporadic Amyo- 3 trophic Lateral Sclerosis
1. Introduction
1.1 within 2–3 45 years, mainly due to a respiratory failure –other publications state up to 5 years
1.2.The exact molecular pathway causing motor neuron degeneration remains still un- 52 known, but as for other ND, is likely to result from a complex interplay between multiple 53 pathogenic cellular mechanisms which may not be mutually exclusive. These include ge- 54 netic factors, excitotoxicity, oxidative stress, mitochondrial dysfunction, impaired axonal 55 transport, neurofilament aggregation, protein aggregation, inflammatory and signalling 56 pathway dysfunctions. - move this passage towards the end of this section
1.3 Plasma was screened by the 86 validated high affinity-based protein technology, called SOMAscan (Multiplexed Proteo- 87 mic technology SOMA logic Inc, Boulder, CO, USA), capable of measuring many human 88 proteins (~ 1300) from a single drop of blood with high sensitivity and specificity. After 89 an accurate bioinformatic analysis, four proteins were selected to be tested as diagnostic 90 markers – this is already a description of the methodology
1.4. Please specify the aims of the study
2. Results
1.2 47 patients with a diagnosis of ALS according to the revised El Escorial criteria [4] 95 were enrolled in the study. 12 patients had a family history of F-ALS with known disease- 96 causing mutations in superoxide dismutase 1 (SOD1), TAR DNA binding protein 97 Int. J. Mol. Sci. 2022, 23, x FOR PEER REVIEW 3 of 24 (TARDBP) or chromosome 9 open reading frame 72 (C9orf72); the remaining 35 were clas- 98 sified as S-ALS. Demographic and clinical features are reported in Table 1 and 2. - then move to the material and methods section
2. 2Characteristics of Control subjects and ALS Patients in first Cohort for SOMAscan Assay. 116 Abbreviations: CTR-Control group; H-CTR-Healthy Control; NoH-CTR-No Healthy Control; SC- 117 sample collection; y-years; SD-standard deviation; F-ALS-familial ALS patients; S-ALS-sporadic 118 ALS patients; FTD-ALS–frontotemporal dementia; ALSci-ALS-cognitive impairment; ALSbi-ALS- 119 behavioral impairment; DNT-data not tested; COPD-Chronic obstructive pulmonary disease. - where is the study group?.Table 1 needs to be rebuilt. There should be a subcategory of sociodemographics etc.
3.3 Tables 1 and 2 can be combined
3. Discussions
1.3ALS is an inevitably fatal disease characterized by the selective degeneration of motor neurons. Moreover, recent studies have described ALS as a highly heterogeneous disease of unknown etiology. This heterogeneity represents a strong limit for the formulation of an early diagnosis and the discovery of effective therapeutic treatments. ALS diagnosis is currently based on clinical criteria and is often formulated by exclusion. Regarding treatment, the drugs approved so far are only able to slow down the disease progression. The discovery of biomarkers that facilitate the diagnosis and/or useful for assessing the progression of the disease and the response to therapies is therefore of considerable importance – this should be in the introduction
4. No conclusions from the research
Author Response

(The authors gave the same response as above.)

Reviewer 3 Report
This manuscript entitled “SOMAscan Proteomics Identifies Plasma Biomarkers with different expression patterns in familial and sporadic Amyotrophic Lateral Sclerosis” by Berrone et al. discussed results of the identification of 42 proteins using SOMAscan involved in Amyotrophic Lateral Sclerosis.
The manuscript is interesting, and in the scope of the journal International Journal of Molecular Sciences. Overall, the manuscript is well written, presented and discussed. However, I recommend some minor revision noted in the following.
Avoid the use of abbreviations in the abstract. It is desirable to define the terms in other parts of the manuscript.
The authors should be improved the introduction. The bibliographic review revealed a lot of shortcomings and reference citation in text.
Author Response

(The authors gave the same response as above.)

Round 2
Reviewer 1 Report
The authors have responded to some extent to several of my concrens, but without addressing a leading concern which was that the statistical approach essentially gives each protein two opportunities to be significant (once with SAM and once with Limma), which invalidates the statistical assumptions of SAM and Limma. In any case, unfortunately, statistical weakness persists in the paper, and extends through to the ELISA results (see below), so even if we might regard the SAM/LImma and Enrichment testing all as exploratory analyses leading up to the ELISA tests, the claims of the paper are not well supported.
With respect to the main claims of the title/abstract and S-ALS v F-ALS:
The F-ALS group is dominated by C9orf patients (6 of 12 in cohort 2), so it is impossible to say whether observed significant differences in the ELISA data are due to the Familial v Sporadic status or due to C9orf v others. In addition to this, although p<0.05 has been observed for several comparisons, it is clear that the ranges of datapoints overlap considerably: as biological measures these may be of interest mechanistically, but as biomarker candidates they are weak. Furthermore, even at the exploratory stage, the heatmap analysis did not show distinction of F-ALS v S-ALS. Finally, the AUC results are underwhelming, and it is not clear why the authors chose S-ALS v Healthy for some combinations and F-ALS v Healthy for other combinations. The main claims of the paper - those of identifying promising biomarkers are not supported by the data, neither at the exploratory stage nor at the ELISA confirmatory stage.
In general, the paper might be better presented as a simple SOMAscan proteomics analysis of plasma proteins in ALS, without making any claims about potential biomarkers (which are not well supported by the data).
Other points/suggestions for the authors:
Line 138-139: "When the 12 proteins resulting from both methods were used only, no clear separation was obtained (data not shown)." Given the heatmap shown, this seems very unlikely. One would expect, strongly, to see some separation between the groups - maybe not perfect, but still obvious. It should be noted that the clustering shown in figure 1 is already not perfect itself: the dendrogram shows that the five ALS samples ordered from A3-1 to B6-1 occupy the same branch of the dendrogram as the Healthy samples. The appearance of clean separation is dependent on the orienttion of the branch, and would disappear if the branch was flipped over (for example, sample A3-1 is more similar to sample C4-1 than it is to sample B7-1, despite that B7-1 and A3-1 are ordered adjacent to each other on the heatmap).
Lines 136-138: "HC on the 24 samples using the deregulated proteins by at least one of the two methods revealed excellent discrimination between ALS patients and H-CTR samples, as shown in Figure 1". As discussed above, it is a bit misleading to describe the discrimination as 'excellent'. the word 'excellent' should be replaced with 'some'.
Line 465: there is a mistake - the p-value cannot be less than 0.
Line 294-296: It is very misleading to suggest that Tusher and Tibshirani (writing their paper before SOMAscan was in use) added a constant to the denominator in order to solve the problem of *proteins* with low expression levels. The authors should rewrite this to make it clear that SAM was developed for gene expression microarrays, and is being applied here to a different data type.
Figure 6: The figure would be more informative if all data-points were overlaid on the box-plots, not just those lying outside the quartile ranges.
Line 37: "validated by ELISA assays in a larger second cohort of patients" > should be "validated by ELISA assays in a larger overlapping cohort of patients"
Reviewer 2 Report
Thank you very much for making the changes. I have no further comments
Author Response
No comment necessary for Reviewer 2
Reviewer 3 Report
The author has made corresponding modifications, and I suggest accepting the manuscript.
Author Response
No comment necessary for Reviewer 3